# Learning with Algorithmic Supervision via Continuous Relaxations

**Felix Petersen**
University of Konstanz
felix.petersen@uni.kn

**Christian Borgelt**
University of Salzburg
christian@borgelt.net

**Hilde Kuehne**
University of Frankfurt
MIT-IBM Watson AI Lab
kuehne@uni-frankfurt.de

**Oliver Deussen**
University of Konstanz
oliver.deussen@uni.kn

## Abstract

The integration of algorithmic components into neural architectures has gained increased attention recently, as it allows training neural networks with new forms of supervision such as ordering constraints or silhouettes instead of using ground truth labels. Many approaches in the field focus on the continuous relaxation of a specific task and show promising results in this context. But the focus on single tasks also limits the applicability of the proposed concepts to a narrow range of applications. In this work, we build on those ideas to propose an approach that allows to integrate algorithms into end-to-end trainable neural network architectures based on a general approximation of discrete conditions. To this end, we relax these conditions in control structures such as conditional statements, loops, and indexing, so that resulting algorithms are smoothly differentiable. To obtain meaningful gradients, each relevant variable is perturbed via logistic distributions and the expectation value under this perturbation is approximated. We evaluate the proposed continuous relaxation model on four challenging tasks and show that it can keep up with relaxations specifically designed for each individual task.

## 1 Introduction

Artificial Neural Networks have shown their ability to solve various problems, ranging from classical tasks in computer science such as machine translation [1] and object detection [2] to many other topics in science such as, e.g., protein folding [3]. Simultaneously, classical algorithms exist, which typically solve predefined tasks based on a given input and a predefined control structure such as, e.g., sorting, shortest-path computation and many more. Recently, research has started to combine both elements by integrating algorithmic concepts into neural network architectures. Those approaches allow training neural networks with alternative supervision strategies, such as learning 3D representations via a differentiable renderer [4], [5] or training neural networks with ordering information [6], [7]. We unify these alternative supervision strategies, which integrate algorithms into the training objective, as algorithmic supervision:

**Definition 1** (Algorithmic Supervision). *In algorithmic supervision, an algorithm is applied to the predictions of a model and the outputs of the algorithm are supervised. In contrast, in direct supervision, the predictions of a model are directly supervised. This is illustrated in Figure 1.*

In general, to allow for end-to-end training of neural architectures with integrated algorithms, the challenge is to estimate the gradients of a respective algorithm, e.g., by a differentiable approximation.

35th Conference on Neural Information Processing Systems (NeurIPS 2021).

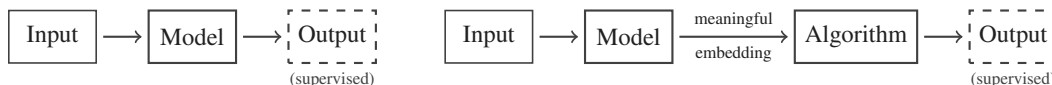

Figure 1: Direct supervision (on the left) in comparison to algorithmic supervision (on the right).

Here, most solutions are tailored to specific problems like, e.g., differentiable sorting or rendering. But also more general frameworks have recently been proposed, which estimate gradients for combinatorial optimizers. Examples for such approaches are the differentiation of black box combinatorial solvers as proposed by Vlastelica *et al.* [8] and the differentiation of optimizers by stochastically perturbing their input as proposed by Berthet *et al.* [9]. Both approaches focus on the problem that it is necessary to estimate the gradients of an algorithm to allow for an end-to-end training of neural architectures including it. To address this issue, Vlastelica *et al.* [8] estimate the gradients of an optimizer by a one-step linearization method that takes the gradients with respect to the optimizer's output into account, which allows to integrate any off-the-shelf combinatorial optimizer. Berthet *et al.* [9] estimate the gradients by perturbing the input to a discrete solver by random noise.

In this context, we propose an approach for making algorithms differentiable and estimating their gradients. Specifically, we propose continuous relaxations of different algorithmic concepts such as comparators, conditional statements, bounded and unbounded loops, and indexing. For this, we perturb those variables in a discrete algorithm by logistic distributions, for which we want to compute a gradient. This allows us to estimate the expected value of an algorithm's output sampling-free and in closed form, e.g., compared to methods approximating the distributions via Monte-Carlo sampling methods (e.g., [9]). To keep the computation feasible, we approximate the expectation value by merging computation paths after each conditional block in sequences of conditional blocks. For nested conditional blocks, we compute the exact expectation value without merging conditional cases. This trade-off allows merging paths in regular intervals, and thus alleviates the need to keep track of all possible path combinations. As we model perturbations of variables when they are accessed, all distributions are independent, which contrasts the case of modeling input perturbations, where all computation paths would have to be handled separately.

To demonstrate the practical aspects, we apply the proposed approach in the context of four tasks, that make use of algorithmic supervision to train a neural network, namely sorting supervision [6], [7], [10], shortest-path supervision [8], [9], silhouette supervision (differentiable rendering) [4], [5], [11], and finally Levenshtein distance supervision. While the first three setups are based on existing benchmarks in the field, we introduce the fourth experiment to show the application of this idea to a new algorithmic task in the field of differentiable dynamic programming. In the latter, the Levenshtein distance between two concatenated character sequences from the EMNIST data set [12] is supervised while the individual letters remain unsupervised. We show that the proposed system is able to outperform current state-of-the-art methods on sorting supervision, while it performs comparable on shortest path supervision and silhouette supervision.

## 2 Related Work

We cover the field of training networks with algorithmic supervision. While some of these works cover single algorithmic concepts, e.g., sorting [6], [7], [13] or rendering [4], [5], [14], others have covered wider areas such as dynamic programming [15], [16] or gradient estimation for general optimizers [8], [9], [17], [18]. Paulus *et al.* [19] employ the Gumbel-Softmax [20] distribution to estimate gradients for concepts such as subset selection. Xie *et al.* [21] and Blondel *et al.* [13] employ differentiable sorting methods for top-$k$ classification. Cuturi *et al.* [22] and Blondel *et al.* [23] present approaches for differentiable dynamic time warping. In machine translation, Bahdanau *et al.* [24] investigate soft alignments, while Collobert *et al.* [25] present a fully differentiable beam sorting algorithm. In speech recognition, Bahdanau *et al.* [26] propose differentiable surrogate task losses for sequence prediction tasks. Shen *et al.* [27] propose a differentiable neural parsing networks using the stick breaking process. Dasgupta *et al.* [28] model box parameters with Gumbel distributions to obtain soft differentiable box embeddings.

In the following, we focus on those works that provide relaxations for tasks considered in this work. We close the overview with a review of some early works on smooth program interpretation.

**Sorting Supervision**   The task of training neural networks with sorting supervision has been first proposed by Grover *et al.* [6]. In this case, a set four-digit numbers based on concatenated MNIST digits [29] is given, and the task is to find an order-preserving mapping from the images to scalars. Here, a CNN should predict a scalar for each of $n$ four-digit numbers such that their order is preserved among the predicted scalars. For training, only sorting supervision in the form of the ground truth order of input images is given, while their absolute values remain unsupervised. Grover *et al.* [6] address this task by relaxing the permutation matrices to double stochastic matrices. Cuturi *et al.* [7] pick up this on benchmark and propose a differentiable proxy by approximating the sorting problem with a regularizing optimal transport algorithm.

**Silhouette Supervision**   A line of work in computer vision is differentiable renderers for 3D-unsupervised 3D-reconstruction. Here, recent approaches have proposed differentiable renderers for 3D mesh reconstruction with only 2D silhouette supervision [4], [5] on 13 classes of the ShapeNet data set [30]. Kato *et al.* [5] propose a renderer where surrogate gradients of rasterization are approximated to perform 3D mesh reconstruction via silhouette supervision as well as 3D style transfer. Liu *et al.* [4] propose a differentiable renderer without surrogate gradients by using a differentiable aggregating process and apply it to 3D mesh reconstruction as well as pose / shape optimization.

**Shortest-Path Supervision**   Two works, proposed by Vlastelica *et al.* [8] and Berthet *et al.* [9] presented methods for estimating gradients for off-the-shelf optimizers. In this context, Vlastelica *et al.* [8] proposed an experiment of shortest-path supervision, where an image of a terrain is given with different costs for each type of terrain, and the task is to predict the shortest path from the top-left to the opposite corner. Integrating a shortest-path algorithm into a neural architecture lets the neural network produce a cost embedding of the terrain which the shortest-path algorithm uses for predicting the shortest path. Vlastelica *et al.* [8] tackle this task by finding a linearization of the Dijkstra algorithm [31], which they can differentiate. Berthet *et al.* [9] take up this problem and produce gradient estimates for the Dijkstra algorithm by stochastically perturbing the inputs to the shortest-path optimization problem.

**Smooth Interpretation**   In another line of work, in the field of computer-aided verification, Chaudhuri *et al.* [32], [33] propose a program smoothing method based on randomly perturbing the inputs of a program by a Gaussian distribution. Here, an initial Gaussian perturbation is propagated through program transformations, and a final distribution over perturbed outputs is approximated via a mixture of Gaussians. The smooth function is then optimized using the gradient-free Nelder-Mead optimization method. The main differences to our method are that we perturb all relevant variables (and not the inputs) with logistic distributions and use this for gradient based optimization.

## 3   Method

To continuously relax algorithms and, thus, make them differentiable, we relax all values with respect to which we want to differentiate into logistic distributions. We choose the logistic distribution as it provides two distinctive properties that make it especially suited for the task at hand: (1) logistic distributions have heavier tails than normal distribution, which allows for larger probability mass and thus larger gradients when two compared values are further away from each other. (2) the cumulative density function (CDF) of the logistic distribution is the logistic sigmoid function, which can be computed analytically, and its gradient is easily computable. This contrasts the CDF of the normal distribution, which has no closed form and is commonly approximated via a polynomial [34].

Specifically, we relax any value $x$, for which we want to compute gradients, by perturbing it into a logistic random variable $\tilde{x} \sim \text{Logistic}(x, 1/\beta)$, where $\beta$ is the inverse temperature parameter such that for $\beta \to \infty : \tilde{x} = x$. Based on this, we can relax a discrete conditional statement, e.g., based on the condition $x < c$ with constant $c \in \mathbb{R}$, as follows:

$$\big[y \text{ if } \tilde{x} < c \text{ else } z\big] \equiv \int_{-\infty}^{c} f_{\log}(t; x, 1/\beta) \cdot y \, \mathrm{d}t \; + \int_{c}^{\infty} f_{\log}(t; x, 1/\beta) \cdot z \, \mathrm{d}t \quad (1)$$

$$= \quad F_{\log}(c; x, 1/\beta) \cdot y \quad + \; (1 - F_{\log}(c; x, 1/\beta)) \cdot z \quad (2)$$

$$= \quad \sigma(c - x) \cdot y \quad + \quad (1 - \sigma(c - x)) \cdot z \quad (3)$$

where $\sigma$ is the logistic (sigmoid) function $\sigma(x) = 1/(1 + e^{-x\beta})$. In this example, as $x$ increases, the result smoothly transitions from $y$ to $z$. Thus, the derivative of the result wrt. $x$ is defined as

$$\frac{\partial}{\partial x}\big[y \texttt{ if } \tilde{x} < c \texttt{ else } z\big] = \frac{\partial}{\partial x}\left(y \cdot \sigma(c - x) + z \cdot (1 - \sigma(c - x))\right) \tag{4}$$

$$= (z - y) \cdot \sigma(c - x) \cdot (1 - \sigma(c - x)) \tag{5}$$

Hence, the gradient descent method can influence the condition $(\tilde{x} < c)$ to hold if the `if` case reduces the loss, or influence the condition to fail if the `else` case reduces the loss function.

In this example, $y$ and $z$ may not only be scalar values but also results of algorithms or parts of an algorithm themselves. This introduces a recursive formalism of relaxed program flow, where parts of an algorithm are combined via a convex combination:

$$\big[f(s) \texttt{ if } a < b \texttt{ else } g(s)\big] \equiv \sigma(b - a) \cdot f(s) + (1 - \sigma(b - a)) \cdot g(s) \tag{6}$$

where $f$ and $g$ denote functions, algorithms, or sequences of statements that operate on the set of all variables $s$ via call-by-value and return the set of all variables $s$. The result of this may either overwrite the set of all variables ($s := [...]$) or be used in a nested conditional statement.

After introducing `if-else` statements above, we extend the idea to loops, which extends the formalism of relaxed program flow to Turing-completeness. In fixed loops, i.e., loops with a predefined number of iterations, since there is only a single computation path, no relaxation is necessary, and, thus, fixed loops can be handled by unrolling.

The more complex case is conditional unbounded loops, i.e., `While` loops, which are executed as long as a condition holds. For that, let $(s_i)_{i \in \mathbb{N}}$ be the sequence of all variables after applying $i$ times the content of a loop. That is, $s_0 = s$ for an initial set of all variables $s$, and $s_i = f(s_{i-1})$, where $f$ is the content of a loop, i.e., a function, an algorithm, or sequence of statements. Let $a, b$ denote accessing variables of $s$, i.e., $s[a], s[b]$, respectively. By recursively applying the rule for `if-else` statements, we obtain the following rule for unbounded loops:

$$\big[\texttt{while } a < b \texttt{ do } s := f(s)\big] \equiv \sum_{i=0}^{\infty} \underbrace{\textstyle\prod_{j=0}^{i}(\sigma(b_j - a_j))}_{(a)} \cdot \underbrace{(1 - \sigma(b_{i+1} - a_{i+1}))}_{(b)} \cdot s_i \tag{7}$$

Here, (a) is the probability that the $i$th iteration is reached and (b) is the probability that there are no more than $i$ iterations. Together, (a) and (b) is the probability that there are exactly $i$ iterations weighing the state of all variables after applying $i$ times $f$, which is $s_i$. Computationally, we evaluate the infinite series until the probability of execution (a) becomes numerically negligible or a predefined maximum number of iterations has been reached. Again, the result may either overwrite the set of all variables ($s := [...]$) or be used in a nested conditional statement.

**Complexity and Merging of Paths**   To compute the exact expectation value of an algorithm under logistic perturbation of its variables, all computation paths would have to be evaluated separately to account for dependencies. However, this would result in an exponential complexity. Therefore, we compute the exact expectation value for nested conditional blocks, but for sequential conditional blocks we merge the computation paths at the end of each block. This allows for a linear complexity in the number of sequential conditional blocks and an exponential complexity only in the largest depth of nested conditional blocks. Note that the number of sequential conditional blocks is usually much larger than the depth of conditional blocks, e.g., hundreds or thousands of sequential blocks and a maximum depth of just $2 - 5$ in our experiments. An example for a dependency is the expression $\big[a := (f(x) \texttt{ if } i < 0 \texttt{ else } g(x)); \; b := (0 \texttt{ if } a < 0 \texttt{ else } a^2)\big]$, which contains a dependence between the two sequential conditional blocks, which introduces the error in our approximation. In general, our formalism also supports modeling dependencies between sequential conditional blocks, however practically, it might become intractable for entire algorithms. Also, it is possible to consider relevant dependencies explicitly if an algorithm relies on specific dependencies.

**Perturbations of Variables vs. Perturbations of Inputs**   Note that modeling the perturbation of variables is different from modeling the perturbation of the inputs. A condition, where the difference becomes clear, is, e.g., $(\tilde{x} < x)$. When modeling input perturbations, the condition would have a strong implicit conditional dependency and evaluate to a $0\%$ probability. However, in this work, we do *not* model perturbations of the inputs, but instead model perturbations of variables each time

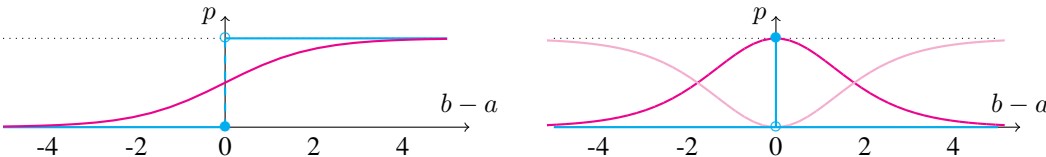

Figure 2: Left: Hard decision boundary (cyan) and probability under logistic perturbation (magenta). Right: Hard equality (cyan), relaxed equality (magenta), and relaxed inequality (light magenta).

they are accessed such that accessing a variable twice is independently identically distributed (iid). Therefore, $(\tilde{x} < x)$ evaluates to a $50\%$ probability. To minimize the approximation error of the relaxation, only those variables should be relaxed for which a gradient is required.

### 3.1 Relaxed Comparators

So far, we have only considered the comparator $<$. $>$ follows by swapping the arguments:

$$\mathbb{P}\left[a < b\right] \equiv \sigma(b - a) \qquad\qquad \mathbb{P}\left[a > b\right] \equiv \sigma(a - b) \qquad (8)$$

**Relaxed Equality** For the equality operator $=$, we consider two distributions $\tilde{a} \sim \mathrm{Logistic}(a, 1/\beta)$ and $\tilde{b} \sim \mathrm{Logistic}(b, 1/\beta)$, which we want to check for similarity / equality. Given value $a$, we compute the likelihood that $a$ is a sample from $\tilde{b}$ rather than $\tilde{a}$. If $a$ is equally likely to be drawn from $\tilde{a}$ and $\tilde{b}$, $\tilde{a}$ and $\tilde{b}$ are equal. If $a$ is unlikely to be drawn from $\tilde{b}$, $\tilde{a}$ and $\tilde{b}$ are unequal. To compute whether it is equally likely for $a$ to be drawn from $\tilde{a}$ and $\tilde{b}$, we take the ratio between the probability that $a$ is from $\tilde{b}$ ($f_{\log}(a; b, 1/\beta)$) and the probability that $a$ is from $\tilde{a}$ ($f_{\log}(a; a, 1/\beta)$):

$$\mathbb{P}\left[a = b\right] \equiv \frac{f_{\log}(a; b, 1/\beta)}{f_{\log}(a; a, 1/\beta)} = \frac{f_{\log}(b; a, 1/\beta)}{f_{\log}(b; b, 1/\beta)} = \mathrm{sech}^2\left(\frac{b - a}{2/\beta}\right) \qquad (9)$$

These relaxed comparators are displayed in Fig. 2. To compute probabilities of conjunction (i.e., and) or of disjunction (i.e., or), we use the product or probabilistic sum, respectively. This corresponds to an intersection / union of independent events. An alternative derivation for Eq. 9 is the normalized conjunction of $\neg(a < b)$ and $\neg(a > b)$.

**Relaxed Maximum** To compare more than two elements and relax the $\arg\max$ function, we use the multinomial logistic distribution, which is also known as the softmax distribution.

$$\mathbb{P}(i = \arg\max_j X_j) = \frac{e^{X_i \beta}}{\sum_j e^{X_j \beta}} \qquad (10)$$

To relax the $\max$ operation, we use the product of $\arg\max$ / softmax and the respective vector.

**Comparing Categorical Variables** To compare a categorical probability distribution $X \in [0, 1]^n$ with a categorical probability distribution $Y$, we consider two scenarios: If $Y \in \{0, 1\}^n$, i.e., $Y$ is one-hot, we can use the inner product of $X$ and $Y$ to obtain their joint probability. However, if $Y \notin \{0, 1\}^n$, i.e., $Y$ is not one-hot, even if $X = Y$ the inner product can not be 1, but a probability of 1 would be desirable if $X = Y$. Therefore, we ($L_2$) normalize $X$ and $Y$ before taking the inner product, which corresponds to the cosine similarity. An example for the application of comparing categorical probability distributions is shown in the Levenshtein distance experiments in Section 4.4.

### 3.2 Relaxed Indexing

As vectors, arrays, and tensors are essential for algorithms and machine learning, we also formalize relaxed indexing. For this, we introduce real-valued indexing and categorical indexing.

**Real-Valued Indexing** In a relaxed algorithm, indices may be drawn from the set of real numbers as they may be a convex combination of computations or a real-valued input. This poses a challenge since it requires interpolating between multiple values. The direct approach would be grid sampling

with bilinear or bicubic interpolation to interpolate values. For example, Neural Turing Machines use linear interpolation for real-valued indexing [35]. However, in bilinear or bicubic interpolation, relationships exceeding the direct (or next) neighbors in the array are not modeled, and they also do not model logistic perturbations. Therefore, we index an $n$-dimensional tensor $\mathbf{A}$ with values $\boldsymbol{i} \in \mathbb{R}^n$ via logistic perturbation by applying a convolution with a logistic filter $g$ and obtain the result as $(g * \mathbf{A})(\boldsymbol{i})$. The convolution of a tensor $\mathbf{A}$ with a logistic filter $g$ (not to be confused with discrete-discrete convolution in neural networks) yields a function that is evaluated at point $\boldsymbol{i} \in \mathbb{R}^n$.

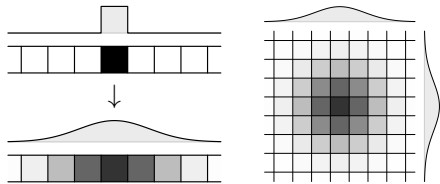

Figure 3: Real-valued indexing: the gray-value represents the extent to which each value is used for indexing. Left: 1D integer-valued hard indexing compared to real-valued indexing. Right: 2D real-valued indexing.

We choose the logistic filter over bilinear and bicubic filters because we model logistic perturbation, and additionally because bilinear and bicubic filters only have compact support whereas the logistic filter provides infinite support. This allows modeling relationships beyond the next neighbors and is more flexible as the inverse temperature $\beta$ can be tuned for the respective application. For stability, we normalize the coefficients used to interpolate the respective indexed values such that they sum up to one: instead of computing $(g * \mathbf{A})(\boldsymbol{i}) = \sum_{\boldsymbol{j}} g(\boldsymbol{j} - \boldsymbol{i}) \mathbf{A}_{\boldsymbol{j}}$, we compute $\sum_{\boldsymbol{j}} g(\boldsymbol{j} - \boldsymbol{i}) \mathbf{A}_{\boldsymbol{j}} / (\sum_{\boldsymbol{j}} g(\boldsymbol{j} - \boldsymbol{i}))$ where $\boldsymbol{j}$ are all valid indices for the tensor $\mathbf{A}$. To prevent optimization algorithms from exploiting aliasing effects, we divide the coefficients by their sum only in the forward pass, but ignore this during the backward pass (computation of the gradient). Real-valued indexing is displayed in Fig. 3, which compares it to hard indexing.

**Relaxed Categorical Indexing** If a categorical probability distribution over indices is given, e.g., computed by $\mathrm{argmax}$ or its relaxation $\mathrm{softmax}$, categorical indexing can be used. Here, the marginal categorical distribution is used as weights for indexing a tensor.

Note that real-valued indexing assumes that the indexed array follows a semantic order such as time series, an image, or the position in a grid. If, in contrast, the array contains categorical information such as the nodes of graphs, values should be indexed with categorical indexing as their neighborhood is arbitrary.

### 3.3 Complexity and Runtime Considerations

In terms of runtime, one has to note that runtime optimized algorithms (e.g., Dijkstra) usually do not improve the runtime for the relaxation, because for the proposed continuous relaxations all cases in an algorithm have to be executed. Thus, an additional condition to reduce the computational cost does not improve runtime because both (all) cases are executed. Instead, it becomes beneficial if an algorithm solves a problem in a rather fixed execution order. On the other hand, optimizing an algorithm with respect to runtime leads to interpolations between the fastest execution paths. This optimization cannot improve the gradients but rather degrades them as it is an additional approximation and produces gradients with respect to runtime heuristics. For example, when relaxing the Dijkstra shortest-path algorithm, there is an interpolation between different orders of visiting nodes, which is the heuristic that makes Dijkstra fast. However, if we have to follow all paths anyway (to compute the relaxation), it can lead to a combinatorial explosion. In addition, by interpolating between those alternative orders, a large amount of uncertainty is introduced, and the gradients will also depend on the orders of visiting nodes, both of which are undesirable. Further, algorithms with rather strict execution can be executed in parallel on GPUs such that they can be faster than runtime-optimized sequential algorithms on CPUs. Therefore, we prefer simple algorithms with a largely fixed execution structure and without runtime optimizations from both a runtime and gradient quality perspective.

## 4 Experiments

We evaluate the proposed approach on various algorithmic supervision experiments, an overview of which is depicted in Fig. 4. For each experiment, we first briefly describe the task and the respective input data as well as the relaxed algorithm that is used. To allow an application to various tasks, we need to find a suitable inverse temperature $\beta$ for each algorithm. As a heuristic, we start with $\beta = \sqrt{k}$ where $k$ is the number of occurrences of relaxed variables in the algorithm, which is a balance between

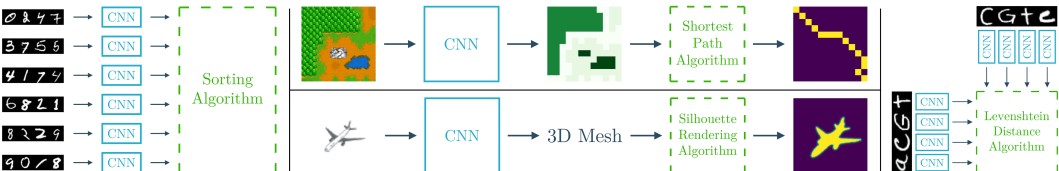

Figure 4: Overview of our algorithmic supervision experiments: Sorting Supervision (left), Shortest-Path Supervision (top), Silhouette Supervision (bottom), and Editing Distance Supervision (right).

a good approximation and sufficient relaxation for good gradient estimates. For each task, we tune this parameter on a validation set. Pseudo-code for all algorithms as well as additional information on the relaxation and the inverse temperature parameter can be found in the supplementary material. The implementation of this work including a high-level PyTorch [36] library for automatic continuous relaxation of algorithms is openly available at github.com/Felix-Petersen/algovision.

## 4.1 Sorting Supervision

In the sorting supervision experiment, a set of four-digit numbers based on concatenated MNIST digits [29] is given, and the task is to find an order-preserving mapping from the images to scalars. A CNN learns to predict a scalar for each of $n$ four-digit numbers such that their order is preserved among the predicted scalars. For this, only sorting supervision in form of the ground truth order of input images is given, while their absolute values remain unsupervised. This follows the protocol of Grover *et al.* [6] and Cuturi *et al.* [7]. An example for a concatenated MNIST image is $8\,9\,2\,7$ .

Using our method, we relax the well established stable sorting algorithm Bubble sort [37], which works by iteratively going through a list and swapping any two adjacent elements if they are not in the correct order until there are no more swap operations in one iteration. We include a variable that keeps track of whether the input sequence is in the correct order by setting it to true if a swap operation occurs. Due to the relaxation, this variable is a floating-point number between $0$ and $1$ corresponding to the probability that the predictions are sorted correctly (under perturbation of variables). We use this probability as the loss function. This variable equals the probability that no swap operation was necessary, and thus $\mathcal{L} = 1 - \prod_{p \in P}(1 - p)$ for probabilities $p$ of each potential swap $p \in P$. By that, the training objective becomes for the input sequence to be sorted and, thus, that the scores predicted by the neural network correspond to the supervised partial order. In fact, the number of swaps in bubble sort corresponds to the Kendall's $\tau$ coefficient, which indicates to which degree a sequence is sorted. Note that, e.g., QuickSort does not have this property as it performs swaps even if the sequence is already sorted.

We emphasize that the task of the trained neural network is *not* to sort a sequence but instead to predict a score for each element such that the ordering / ranking corresponds to the supervised ordering / ranking. While the relaxed algorithm can sort the inputs correctly, at evaluation time, following the setup of [6] and [7] we use an `argsort` method to test whether the outputs produced by the neural network are in accordance with the ground truth partial order. We use the same network architecture as [6] and [7]. Here, we only optimize the inverse temperature for $n = 5$, resulting in $\beta = 8$, and fix this for all other $n$ . For training, we use the Adam optimizer [38] with a learning rate of $10^{-4}$ for between $1.5 \cdot 10^5$ and $1.5 \cdot 10^6$ iterations.

We evaluate our method against state-of-the-art hand-crafted relaxations of the sorting operation using the same network architecture and evaluation metrics as Grover *et al.* [6] and Cuturi *et al.* [7]. As displayed in Tab. 1, our general formulation outperforms state-of-the-art hand-crafted relaxations of the sorting operation for sorting supervision.

Table 1: Results for the 4-digit MNIST sorting task, averaged over 10 runs. Baselines as reported by Cuturi *et al.* [7]. Trained and evaluated on sets of $n$ elements, the displayed metrics are exact matches (and element-wise correct ranks).

| Method | $n = 3$ | $n = 5$ | $n = 7$ |
|---|---|---|---|
| Stoch. NeuralSort [6] | 0.920 (0.946) | 0.790 (0.907) | 0.636 (0.873) |
| Det. NeuralSort [6] | 0.919 (0.945) | 0.777 (0.901) | 0.610 (0.862) |
| Optimal Transport [7] | 0.928 (0.950) | 0.811 (0.917) | 0.656 (0.882) |
| Relaxed Bubble Sort | **0.944 (0.961)** | **0.842 (0.930)** | **0.707 (0.898)** |

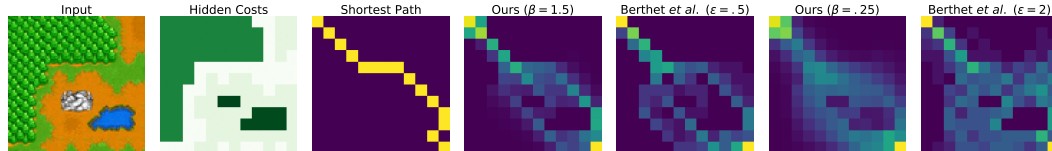

Figure 5: (From left to right.) Example input image of the Warcraft terrain data set with the hidden cost matrix and the resulting shortest path. Shortest paths relaxed with the proposed method for $\beta \in \{1.5, 0.25\}$, which correspond to the perturbed paths by Berthet *et al.* [9] with $\epsilon \in \{0.5, 2.0\}$.

## 4.2 Shortest-Path Supervision

For shortest-path supervision on 2D terrains, we follow the setup by Vlastelica *et al.* [8] and Berthet *et al.* [9] and use the data set of $10\,000$ patches of Warcraft terrains of size $96 \times 96$ representing terrain grids of size $12 \times 12$. Given an image of a terrain (e.g., Fig. 5 first), the goal is to predict the shortest path (e.g., Fig. 5 third) according to a hidden cost matrix (e.g., Fig. 5 second). For this, $12 \times 12$ binary matrices of the shortest-path are supervised, while the hidden cost matrix is used only to determine the shortest path. In their works, Vlastelica *et al.* [8] and Berthet *et al.* [9] show that integrating and differentiating a shortest-path algorithm can improve the results by allowing the neural network to predict a cost matrix from which the shortest path can be computed via the algorithm. This performs significantly better than a ResNet baseline where the shortest paths have to be predicted by the neural network alone (see Tab. 2).

For this task, we relax the Bellman-Ford algorithm [39] with 8-neighborhood, node weights, and path reconstruction, details on which are given in the supplementary material. For the loss between the supervised shortest paths and the paths produced by the relaxed Bellman-Ford algorithm, we use the $\ell^2$ loss. To illustrate the shortest paths created by our method and to compare them to those created through the Perturbed Optimizers by Berthet *et al.* [9], we display examples of back traced shortest paths for two inverse temperatures in Fig. 5 (center to right).

Table 2: Results for the Warcraft shortest-path task using shortest-path supervision, averaged over 10 runs. Reported are exact matches (EM) as well as the ratio between the cost of the predicted path and the cost of the optimal shortest path.

| Method | EM | cost ratio |
|---|---|---|
| ResNet Baseline [8] | 40.2% | 1.01530 |
| Black-Box Loss [8] | 86.6% | 1.00026 |
| Perturbed Opt. [9] | **91.7%** | 1.00042 |
| Relaxed Shortest-Path | 88.4% | **1.00014** |

We use the same ResNet network architecture as Vlastelica *et al.* [8] and Berthet *et al.* [9] and also train for $50$ epochs with a batch size of $70$. We use an inverse temperature of $\beta = 25$.

As shown in Tab. 2, our relaxation outperforms all baselines on the cost ratio of the predicted to the optimal shortest path and achieves the second-best result on the shortest path exact match accuracy after the perturbed optimizers [9].

## 4.3 Silhouette Supervision

Reconstructing a 3D model from a single 2D image is an important task in computer vision. Recent works [4], [5] have employed differentiable renderers to train a 3D reconstruction network by giving feedback on whether the silhouettes of a reconstruction match the input image. Specifically, recent works have been benchmarked [4], [5], [11] on a data set of 13 object classes from ShapeNet [30] that have been rendered from $24$ azimuths at a resolution of $64 \times 64$ [5]. For learning with silhouette supervision, a single image is processed by a neural network, which returns a 3D mesh. The silhouette of this mesh is rendered from two view-points by a differentiable renderer and the intersection-over-union between the rendered and predicted meshes is used as a training objective to update the neural network [4], [5]. For training, we also use the same neural network architecture as Kato *et al.* [5] as well as Liu *et al.* [4]. Note that, while renderers can be able to render a full RGB image, in these experiments, only the silhouette is used for supervision of the 3D geometry reconstruction. Specifically, public implementations of [4], [5] only use silhouette supervision.

Table 3: Single-view 3D reconstruction results using silhouette supervision. Reported is the 3D IoU.

| Method | Airplane | Bench | Dresser | Car | Chair | Display | Lamp | Speaker | Rifle | Sofa | Table | Phone | Vessel | *Mean* |
|---|---|---|---|---|---|---|---|---|---|---|---|---|---|---|
| **With a batch size of 64** | | | | | | | | | | | | | | |
| Yan *et al.* [11] (retrieval) | 0.5564 | 0.4875 | 0.5713 | 0.6519 | 0.3512 | 0.3958 | 0.2905 | 0.4600 | 0.5133 | 0.5314 | 0.3097 | 0.6696 | 0.4078 | 0.4766 |
| Yan *et al.* [11] (voxel) | 0.5556 | 0.4924 | 0.6823 | 0.7123 | 0.4494 | 0.5395 | 0.4223 | 0.5868 | 0.5987 | 0.6221 | 0.4938 | 0.7504 | 0.5507 | 0.5736 |
| Kato *et al.* [5] (NMR) | 0.6172 | 0.4998 | 0.7143 | 0.7095 | 0.4990 | 0.5831 | 0.4126 | 0.6536 | 0.6322 | 0.6735 | 0.4829 | 0.7777 | 0.5645 | 0.6015 |
| Liu *et al.* [4] (SoftRas) | 0.6419 | 0.5080 | 0.7116 | 0.7697 | 0.5270 | 0.6156 | 0.4628 | 0.6654 | 0.6811 | 0.6878 | 0.4487 | 0.7895 | 0.5953 | 0.6234 |
| **With a batch size of 2** | | | | | | | | | | | | | | |
| Liu *et al.* [4] (SoftRas) | **0.5741** | **0.3746** | 0.6373 | **0.6939** | **0.4220** | 0.5168 | 0.4001 | 0.6068 | **0.6026** | **0.5922** | 0.3712 | **0.7464** | **0.5534** | **0.5455** |
| Relaxed Sil. via Three Edges | 0.5418 | 0.3667 | **0.6626** | 0.6546 | 0.3899 | 0.5229 | 0.4105 | **0.6232** | 0.5497 | 0.5639 | 0.3580 | 0.6609 | 0.5279 | 0.5256 |
| Relaxed Sil. via Euclid. Dist | 0.5399 | 0.3698 | 0.6503 | 0.6524 | 0.4044 | **0.5261** | **0.4247** | 0.6225 | 0.5723 | 0.5643 | **0.3829** | 0.7265 | 0.5180 | 0.5349 |

For this task, we relax two silhouette rendering algorithms. The algorithms rasterize a 3D mesh as follows: For each pixel and for each triangle of the mesh, if a pixel lies inside a triangle, the value of the pixel in the output image is set to 1. The condition of whether a pixel lies inside a triangle is checked in two alternative fashions: (1) by three nested `if` conditions that check on which side of each edge the pixel lies. (2) by checking whether the directed euclidean distance between a pixel and a triangle is positive. Note that, by relaxing these algorithms using our framework, we obtain differentiable renderers very close to Pix2Vex [40] for (1) and SoftRas [4] for (2). Examples of relaxed silhouette renderings and an example image from the data set are displayed in Fig. 6.

As the simple silhouette renderer does not have any optimizations, such as discarding pixels that are far away from a triangle or triangles that are occluded by others, it is not very efficient. Thus, due to limited resources, we are only able to train with a maximum batch size of 2 while previous works used a batch size of 64. Therefore, we reproduce the recent best performing work by Liu *et al.* [4] on a batch size of only 2 to allow for a fair comparison. For directed Euclidean distance, we use an inverse temperature of $\beta = 2\,000$; for three edges, $\beta = 10\,000$.

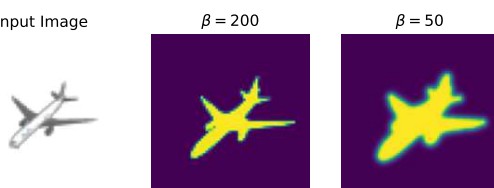

Figure 6: An input image from the data set (left). Silhouette of a prediction rendered with directed Euclidean distance approach for two different inverse temperatures $\beta \in \{200, 50\}$.

We report the average as well as class-wise 3D IoU results in Tab. 3. Our relaxations outperform the retrieval baseline by Yan *et al.* [11] even though we use a batch size of only 2. In direct comparison to the SoftRas renderer at a batch size of 2, our relaxations achieve the best accuracy for 5 out of 13 object classes. However, on average, our methods, although they do not outperform SoftRas, show a comparable performance with a drop of only 1%. It is notable that the directed Euclidean distance approach performs better than the three edges approach. The three edges approach is faster by a factor of 3 because computing the directed Euclidean distance is more expensive than three nested `if` clauses (even in the relaxed case.)

## 4.4  Levenshtein Distance Supervision

In the Levenshtein distance supervision experiment, we supervise a classifier for handwritten EMNIST letters [12] by supervising the editing distance between two handwritten strings of length 32 only. As editing distance, we use the Levenshtein distance LD which is defined as $\mathrm{LD}(a, b) = \mathrm{LD}_{|a|,|b|}(a, b)$:

$$\mathrm{LD}_{i,j}(a,b) = \begin{cases} j & i = 0 \\ i & j = 0 \\ \min \begin{cases} \mathrm{LD}_{i-1,j}(a,b) + 1 \\ \mathrm{LD}_{i,j-1}(a,b) + 1 \\ \mathrm{LD}_{i-1,j-1}(a,b) + \mathbb{1}_{(a_{i-1} \neq b_{j-1})} \end{cases} & \text{else.} \end{cases} \tag{11}$$

We relax the Levenshtein distance using its classic dynamic programming algorithm. An example Levenshtein distance matrix and its relaxation are displayed in Fig. 7.

Table 4: EMNIST classification results with Levenshtein distance supervision averaged over 10 runs.

| Method | | AB | BC | CD | DE | EF | IL | OX | ACGT | OSXL |
|--------|--|----|----|----|----|----|----|----|------|------|
| Baseline | Top-1 acc. | .616 | .651 | .768 | .739 | .701 | .550 | .893 | .403 | .448 |
| | F1-score | .581 | .629 | .759 | .711 | .674 | .490 | .890 | .336 | .384 |
| Relaxed LD | Top-1 acc. | **.671** | **.807** | **.816** | **.833** | **.847** | **.570** | **.960** | **.437** | **.487** |
| | F1-score | **.666** | **.805** | **.815** | **.831** | **.845** | **.539** | **.960** | **.367** | **.404** |

For learning, pairs of strings of images of 32 handwritten characters $a, b$ as well as the ground truth Levenshtein distance $\mathrm{LD}(\mathbf{y}_a, \mathbf{y}_b)$ are given. We sample pairs of strings $a, b$ from an alphabet of 2 or 4 characters. For sampling the second string given the first one, we uniformly choose between two and four insertion and deletion operations. Thus, the average editing distance for strings, which use two different characters, is $4.25$ and for four characters is $5$. We process each letter, using a CNN, which returns a categorical distribution over letters, which is then fed to the algorithm. An example for a pair of strings based on $\{A, C, G, T\}$ is

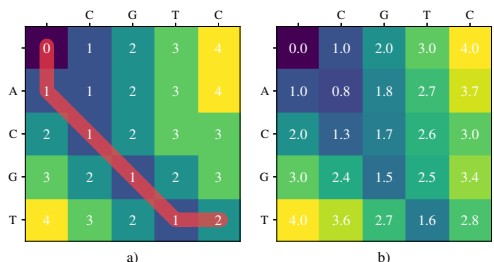

Figure 7: Levenshtein distance matrix. Left: hard matrix and alignment path. Right: relaxed matrix with inverse temperature $\beta = 1.5$.

![string image] and
![string image] . Our training objective is minimizing the $\ell^2$ loss between the predicted distance and the ground truth distance:

$$\mathcal{L} = \left\| \mathrm{LD}\left( (\mathrm{CNN}(a_i))_{i \in \{1..32\}}, (\mathrm{CNN}(b_i))_{i \in \{1..32\}} \right) - \mathrm{LD}(\mathbf{y}_a, \mathbf{y}_b) \right\|_2. \tag{12}$$

For training, we use an inverse temperature of $\beta = 9$ and Adam ($\eta = 10^{-4}$) for $128 - 512$ iterations.

For evaluation, we use Hungarian matching with Top-1 accuracy as well as the F1-score. We compare it against a baseline, which uses the $\ell_1$ distance between encodings instead of the editing distance:

$$\mathcal{L} = \left\| \left\| (\mathrm{CNN}(a_i))_{i \in \{1..32\}} - (\mathrm{CNN}(b_i))_{i \in \{1..32\}} \right\|_1 - \mathrm{LD}(\mathbf{y}_a, \mathbf{y}_b) \right\|_2. \tag{13}$$

Tab. 4 shows that our method consistently outperforms the baseline on both metrics in all cases. The character combinations AB, BC, CD, DE, and EF are a canonical choice for random combinations. The characters IL represent are the hardest combination of two letters as they even get frequently confused by supervised neural networks [12] and can also be indistinguishable for humans. The characters OX represent the easiest case as supervised classifiers can perfectly distinguish them on the test dataset [12]. For two letter combinations, we achieve Top-1 accuracies between $57\%$ (IL) and $96\%$ (OX). Even for four letter combinations (ACGT and OSXL), we achieve Top-1 accuracies of up to $48.7\%$. Note that, as we use strings of length 32, in the Levenshtein algorithm, more than $1\,000$ statements are relaxed.

## 5  Conclusion

We proposed an approach for continuous relaxations of algorithms that allows their integration into end-to-end trainable neural network architectures. For that, we use convex combinations of execution paths of algorithms that are parameterized by smooth functions. We show that our proposed general framework can compete with SOTA continuous relaxations of specific algorithms as well as gradient estimation methods on a variety of algorithmic supervision tasks. Moreover, we show that our formulation successfully relaxes even complex algorithms such as a shortest-path algorithm or a renderer. We hope to inspire the research community to build on our framework to further explore the potential of algorithmic supervision and algorithmically enhanced neural network architectures.

## Acknowledgments and Disclosure of Funding

This work was supported by the IBM-MIT Watson AI Lab, the DFG in the SFB Transregio 161 "Quantitative Methods for Visual Computing" (Project-ID 251654672), the DFG in the Cluster of Excellence EXC 2117 "Centre for the Advanced Study of Collective Behaviour" (Project-ID 390829875), and the Land Salzburg within the WISS 2025 project IDA-Lab (20102-F1901166-KZP and 20204-WISS/225/197-2019).

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
