# Supplementary Material for Learning with Algorithmic Supervision via Continuous Relaxations

**Felix Petersen**     **Christian Borgelt**     **Hilde Kuehne**     **Oliver Deussen**

In the supplementary material, we give implementation details, and present the algorithms.

## A    Implementation Details

### A.1    Sorting Supervision

**Network Architecture**    For comparability to Grover *et al.* [6] and Cuturi *et al.* [7], we use the same network architecture. That is, two convolutional layers with a kernel size of $5 \times 5$, 32 and 64 channels respectively, each followed by a ReLU and MaxPool layer; after flattening, this is followed by a fully connected layer with a size of $64$, a ReLU layer, and a fully connected output layer mapping to a scalar.

### A.2    Shortest-Path Supervision

**Network Architecture**    For comparability to Vlastelica *et al.* [8] and Blondel *et al.* [23], we use the same network architecture. That is, the first five layers of ResNet18 followed by an adaptive max pooling to the size of $12 \times 12$ and an averaging over all features.

**Training**    As in previous works, we train for 50 epochs with batch size 70 and decay the learning rate by $0.1$ after $60\%$ as well as after $80\%$ of training.

### A.3    Silhouette Supervision

**Network Architecture**    For comparability to Liu *et al.* [4], we use the same network architecture. That is, three convolutional layers with a kernel size of $5 \times 5$, 64, 128, and 256 channels respectively, each followed by a ReLU; after flattening, this is followed by 6 ReLU-activated fully connected layers with the following output dimensions: $1024, 1024, 512, 1024, 1024, 642 \times 3$. The $642 \times 3$ elements are interpreted as three dimensional vectors that displace the vertices of a sphere with 642 vertices.

**Training**    We train the Three Edges approach with Adam ($\eta = 5 \cdot 10^{-5}$) for $2.5 \cdot 10^6$ iterations and train the directed Euclidean distance approach with Adam ($\eta = 5 \cdot 10^{-5}$) for $10^6$ iterations. The reason for this is that each of them took around 6 days of training on a single V100 GPU. We decay the learning rate by $0.3$ after $60\%$ as well as after $80\%$ of training.

### A.4    Levenshtein Distance Supervision

**Network Architecture**    The CNN consists of two convolutional layers with a kernel size of 5 and hidden sizes of 32 and 64, each followed by a ReLU, and a max-pooling layer. The convolutional layers are followed by two fully connected layers with a hidden size of 64 and a ReLU activation.

## B  Standard Deviations for Results

Table 5: Sorting Supervision: Standard deviations for Table 1.

| Method | $n = 3$ | $n = 5$ | $n = 7$ |
|---|---|---|---|
| Relaxed Bubble Sort | $0.944 \pm .009$ | $0.842 \pm .012$ | $0.707 \pm .008$ |
| | $(0.961 \pm .006)$ | $(0.930 \pm .005)$ | $(0.898 \pm .003)$ |

Table 6: Shortest-Path Supervision: Standard deviations for Table 2.

| Method | EM | cost ratio |
|---|---|---|
| Black-Box Loss [8] | $86.6\% \pm 0.8\%$ | $1.00026 \pm 0.00005$ |
| Relaxed Shortest-Path | $88.4\% \pm 0.7\%$ | $1.00014 \pm 0.00008$ |

Table 7: Levenshtein Distance Supervision: Standard deviations for Table 4.

| Method | AB | BC | CD | DE | EF | IL | OX | ACGT | OSXL |
|---|---|---|---|---|---|---|---|---|---|
| Baseline | $.616 \pm .041$ | $.651 \pm .099$ | $.768 \pm .062$ | $.739 \pm .107$ | $.701 \pm .097$ | $.550 \pm .039$ | $.893 \pm .088$ | $.403 \pm .065$ | $.448 \pm .059$ |
| | $.581 \pm .059$ | $.629 \pm .120$ | $.759 \pm .073$ | $.711 \pm .151$ | $.674 \pm .124$ | $.490 \pm .095$ | $.890 \pm .094$ | $.336 \pm .084$ | $.384 \pm .078$ |
| Relaxed LD | $.671 \pm .103$ | $.807 \pm .095$ | $.816 \pm .060$ | $.833 \pm .038$ | $.847 \pm .091$ | $.570 \pm .027$ | $.960 \pm .079$ | $.437 \pm .026$ | $.487 \pm .076$ |
| | $.666 \pm .107$ | $.805 \pm .097$ | $.815 \pm .060$ | $.831 \pm .039$ | $.845 \pm .097$ | $.539 \pm .042$ | $.960 \pm .080$ | $.367 \pm .051$ | $.404 \pm .104$ |

## C  Algorithms

### C.1  Sorting Supervision: Bubble Sort

On the left, a Python version reference implementation of bubble sort [37] is displayed. On the right, the relaxed version is displayed.

```
def bubble_sort(A):      bubble_sort = Algorithm(
  n = len(A) - 1            Lambda(lambda A: A.shape[-1] - 1, ['n'])
  swapped = True            Lambda(lambda swapped: 1.)
  while swapped:            While('swapped', Sequence(
    swapped = False           Lambda(lambda swapped: 0),
    for i in range(n):        For('i', 'n', Sequence(
      if A[i] > A[i+1]:         If(GT(lambda A, i: IndexInplace()(A, i),
                                       lambda A, i: IndexInplace()(A, i+1)),
                                  if_true=Sequence(
        a_1 = A[i+1]              Index('a_1', 'A', lambda i: i+1),
        a_2 = A[i]               Index('a_2', 'A', 'i'),
        A[i]   = a_1             IndexAssign('A', 'i', 'a_1'),
        A[i+1] = a_2             IndexAssign('A', lambda i: i+1, 'a_2'),
        swapped = True           Lambda(lambda swapped: 1.),
        loss = 1                 Lambda(lambda loss: 1.) )))),
    n = n - 1               Lambda(lambda n: n - 1)
  return A                ) ) )
```

### C.2  Shortest-Path Supervision: Bellman-Ford

In the following, we provide pseudo-code for the Bellman-Ford algorithm with 8-neighborhood, node weights, and path reconstruction.

```
def shortest_path(cost):
    n = cost.shape[0]
    D[0:n+2, 0:n+2] = INFINITY
    D[1, 1] = 0
    for _ in range(n*n):
        arg_D = arg_minimum_neighbor(D)      # 8-neighborhood
        D = cost + minimum_neighbor(D)
        D[1, 1] = 0

    path[0:n+2, 0:n+2] = 0
    position = n+1, n+1
    while path[1, 1] == 0:
        path[position] = 1
        position = get_next_location(arg_D, position)

    return path
```

For the relaxation, `arg_minimum_neighbor` and `minimum_neighbor` use softmax. Further, for the relaxation, `get_next_location` returns a marginal distribution over all possible positions. An alternative, where `get_next_location` returns a pair of real-valued coordinates is possible, however the quality of the gradients is reduced.

## C.3  Silhouette Supervision: 3D Mesh Renderer

In the following, we provide pseudo-code for the two simple silhouette rendering algorithms that we use.

### C.3.1  Three Edges

```
def silhouette_renderer(triangles, camera_extrinsics, resolution=64):
    triangles = transform_and_projection(triangles, camera_extrinsics)

    image[0:resolution, 0:resolution] = 0
    for p_x in range(resolution):
        for p_y in range(resolution):
            for t in triangles:
                # t.e1, t.e2, t.e3 are the three edges of t
                if directed_dist(t.e1, p_x, p_y) <= 0:
                    if directed_dist(t.e2, p_x, p_y) <= 0:
                        if directed_dist(t.e3, p_x, p_y) <= 0:
                            image[p_x, p_y] = 1
                else:
                    if directed_dist(t.e2, p_x, p_y) > 0:
                        if directed_dist(t.e3, p_x, p_y) > 0:
                            image[p_x, p_y] = 1
    return image
```

### C.3.2  Directed Euclidean Distance

```
def silhouette_renderer(triangles, camera_extrinsics, resolution=64):
    triangles = transform_and_projection(triangles, camera_extrinsics)

    image[0:resolution, 0:resolution] = 0
    for p_x in range(resolution):
        for p_y in range(resolution):
            for t in triangles:
                if directed_euclidean_distance(t, p_x, p_y) <= 0:
                    image[p_x, p_y] = 1
    return image
```

For both algorithms, we parallelize the three loops as they are independent. As for runtime, the Three Edges algorithm is around 3 times faster than the directed euclidean distance algorithm. This is because computing the euclidean distance between a point and a triangle is an expensive operation.

## C.4   Levenshtein Distance Supervision (Dynamic Programming)

Pseudo-code of our implementation of the Levenshtein distance [41] and a simplified code for our framework is displayed below.

```
def levenshtein_distance(s, t):
    n = len(s)
    d[0:n + 1, 0:n + 1] = 0
    for i in range(n):
        d[i + 1, 0] = i + 1
    for j in range(n):
        d[0, j + 1] = j + 1
    for i in range(n):
        for j in range(n):
            if s[i] == t[j]:
                subs_cost = 0
            else:
                subs_cost = 1
            d[i + 1, j + 1] = min(  d[i, j + 1] + 1,
                                    d[i + 1, j] + 1,
                                    d[i, j] + subs_cost )
    return d[n, n]

levenshtein_distance = Algorithm(
  For('i', 'n',
    IndexAssign2D('d', lambda i: [i + 1, i*0], lambda i: i + 1) ),
  For('j', 'n',
    IndexAssign2D('d', lambda j: [i*0, j + 1], lambda j: j + 1) ),
  For('j', 'n',
    For('i', 'n', Sequence(
      If(CatProbEq(lambda s, i: IndexInplace(s, i),
                   lambda t, j: IndexInplace(t, j) ),
        if_true= Lambda(lambda subs_cost: 0),
        if_false=Lambda(lambda subs_cost: 1),
        ),
        IndexAssign2D('d',
                      index=lambda i, j: [i + 1, j + 1],
                      value=lambda d, i, j, subs_cost:
                        Min(d[:, i, j + 1] + 1,
                            d[:, i + 1, j] + 1,
                            d[:, i, j] + subs_cost ) )
    ) )
  )
)
```