# OpenReview forum: "Learning with Algorithmic Supervision via Continuous Relaxations"
_NeurIPS.cc/2021/Conference — NeurIPS 2021 Poster_

### Official Review · Reviewer_nw3A · 2021-07-12

**Rating:** 6
**Confidence:** 4

**Summary:**

The continuous relaxation of certain elements of certain programming primitives have been used in deep learning to incorporate constraints or logic into architectures. Attention, for example, has been viewed as a key-value dictionary lookup operation.
This paper aims to formalise some of these operations in a more general framework.

The main aspect that is considered is the if-else construct in programming. Rewriting the if-else conditionals as an integral over the logistic, they arrive at the classic gating function we often use in deep learning (e.g. GRUs, LSTM). This is extended to while loops, which are repeated applications of the if-else conditional. This particular equation looks like an expectation over the stick-breaking process (https://en.wikipedia.org/wiki/Dirichlet_process#The_stick-breaking_process)

Similar relaxations are then proposed for the equals comparator, with interesting initial assumptions that lead to the use of sech. Relaxed indexing also accounts for cases where the indexing is ordinal.


**Ethical Concerns:**

N/A: Work is largely theoretical

**Limitations And Societal Impact:**


The authors have addressed one particular limitation of such relaxations, which is in combining separate branches.


**Main Review:**

Method and related work

The explanation of their approach to relaxation of these if-else constructs is clear. I am unsure if this has been done before, but I like the generalised framework their approach provides.
However, I think this paper would go over better if it made connections to prior work, and how they are specific cases of the framework provided in this paper. This allows readers who are unfamiliar with the more abstract aspect of the work to ground it in something they may have encountered before, and be able to see it more clearly as a generalisation of something they already know.

An example would be the Neural Turing Machine, and the various gating systems could be reframed as if-else conditions.

Some references of this nature:
If-else:
Graves, Alex, Greg Wayne, and Ivo Danihelka. "Neural turing machines."
Shen, Yikang, et al. "Ordered memory."

Stick-breaking while loop

- Yogatama, Dani, et al. "Memory architectures in recurrent neural network language models."
- Shen, Yikang, et al. "Neural language modeling by jointly learning syntax and lexicon."
- Graves, Alex. "Adaptive computation time for recurrent neural networks."
- Joulin, Armand, and Tomas Mikolov. "Inferring algorithmic patterns with stack-augmented recurrent nets."

Indexing:
- Graves, Alex, Greg Wayne, and Ivo Danihelka. "Neural turing machines."
- Graves, Alex, et al. "A novel connectionist system for unconstrained handwriting recognition."
(attention system over image)

This work also has some connections to other work that deal with relaxation of known algorithms.
Another paper that creates a framework for relaxations of certain structure-based algorithms is:
- Mensch, Arthur, and Mathieu Blondel. "Differentiable dynamic programming for structured prediction and attention."

Specifically for the Eisner algorithm:
- Corro, Caio, and Ivan Titov. "Differentiable perturb-and-parse: Semi-supervised parsing with a structured variational autoencoder."

Experiments

MNIST Sorting Very surprising that relaxed bubble sort works better than the sinkhorn based method.

Warcraft shortest-path: Do you un-relax the algorithm during test time? Using the cost computed with the relaxed approach might result in lower costs.

Might be interesting to check how well do the methods generalise to untrained input sizes (e.g. longer lengths for MNIST, larger maps for shortest-path, ...)


**Time Spent Reviewing:**

3

---

> ### Author Response · Authors · 2021-08-10
> **Response**
>
> Thank you for your helpful and encouraging comments.
>
> **"However, I think this paper would go over better if it made connections to prior work, and how they are specific cases of the framework provided in this paper.":**
> We agree that this an important point which we will include in the related work discussion.
>
> **References:** Thank you for providing all of these interesting additional references. We will add them to the discussion in the related work section.
>
> **Warcraft shortest-path: Do you un-relax the algorithm during test time?:**
> Yes, following the previous works, we use the original / hard / un-relaxed algorithm during test time to avoid biasing the evaluation.
>
> **Might be interesting to check how well do the methods generalise to untrained input sizes (e.g. longer lengths for MNIST, larger maps for shortest-path, ...):**
>
> * For MNIST, we evaluated a model trained for n=3 for predicting rankings of 5 elements. Here we obtain an exact match accuracy of 0.835 (instead of 0.842 for actually training with 5 elements.) Any other input sizes would also be feasible during inference.
> * As for the shortest-path data set, only the 12x12 maps have been released by the authors.

---

> > ### Author Response · Authors · 2021-08-26
> > **Thanks**
> >
> > We thank you for your helpful comments and hope you had a chance to look at our response. Please let us know if we addressed your concerns and questions. In case our responses addressed your concerns, we would appreciate if you would consider increasing the score.

---

> > > ### Comment · Reviewer_nw3A · 2021-08-30
> > > **Thank you for your response**
> > >
> > > I will however maintain the score I gave.

---

### Official Review · Reviewer_YtiF · 2021-07-16

**Rating:** 6
**Confidence:** 3

**Summary:**

The paper proposes an approach for continuous relaxations of algorithms that allows their integration into end-to-end trainable neural network architectures. It shows that the general approach can compete with SOTA continuous relaxations and could relaxes for complex algorithms, e.g., shortest-path algorithm or a renderer. Some experiments show the effectiveness.

**Limitations And Societal Impact:**

I have following comments.
1. Why the proposed method is inferior to Perturbed Opt? It is better to explain this in detail.
2. It is better to estimate the scalability of the proposed theoretically.

**Main Review:**

The paper is well-written. It includes the probabilistic interpretation in Section 3. The two experiments, i.e., shortest-path and rendering are complex, which may testify to the  scalability of the proposed method.

**Time Spent Reviewing:**

4

---

> ### Author Response · Authors · 2021-08-10
> **Response**
>
> Thank you for your helpful and encouraging comments.
>
> **Performance in comparison to Perturbed Opt:** While perturbed opt performs better on one metric (EM), it is not performing as good as our method on the other metric (cost ratio).
> However, the exact match accuracy does have the issue that there can be two or more optimal paths.
> Following the original implementation by Vlastelica et al. [8], we only consider it to be an exact match if the predicted shortest path is the same as the ground truth path, which is only one out of potentially multiple optimal paths.
> Thus, sometimes, a correct prediction is rejected.
> In Table 2, we used the results for Perturbed Opt. from Berthet et al. [9], while we have produced all other results in Table 2 ourselves.
> We are not able to confirm which definition of EM was used in the Perturbed Opt. experiments.
>
>
> **Scalability / Runtime Analysis:**
> We did not report this, but, except for the differentiable renderer, the runtime of the upstream neural network is larger than the runtime of the relaxed algorithm.
> Therefore, the runtime of the relaxed algorithm was a rather minor concern.
> We will include a runtime analysis in the supplementary material.

---

> > ### Author Response · Authors · 2021-08-26
> > **Thanks**
> >
> > We thank you for your helpful comments and hope you had a chance to look at our response. Please let us know if we addressed your concerns and questions. In case our responses addressed your concerns, we would appreciate if you would consider increasing the score.

---

### Official Review · Reviewer_1ez2 · 2021-07-20

**Rating:** 7
**Confidence:** 4

**Summary:**


This paper proposes to integrate discrete algorithmic components into a deep network, or its learning objective, by continuous relaxations of the discrete operations. It also proposes to consider independently the relaxation of each operation and their combinations (what's referred to as "variable perturbations"), which avoids the combinatorial explosion of having to consider all possible paths (which would model higher-order interactions between variables, which are ignored here).


**Limitations And Societal Impact:**


No. The authors could mention cases where they expect their strategy to fail (such as some concrete examples where they expect high-order interactions between variables to play a large role). Since this proposal is fairly application-agnostic, societal impact does not seem to be a primary concern.


**Main Review:**

Although there are many attempts in the literature to propose discrete relaxations for deep networks, these are usually fairly ad-hoc and problem-specific. This proposal focuses on generic strategies for relaxation based on a single core element, the logistic distribution. This means that it is somewhat more formally satisfying than previous proposals. There's also a great focus on avoiding Monte Carlo sampling, which should be very impactful in practice, although this is not explored fully. Thus, I believe that this is a very well-crafted and novel solution to a very general problem.

The writing is of good quality, although there are instances that could be more clear, and especially a few where a formula would help make things more concrete. The related work section is relatively thorough and descriptive (except for one omission), and the experiments are well described and cover a wide array of areas, although each experiment in isolation is a bit small-scale. I did not identify any problems with the formulation, which was easy enough to follow.

In the related work, one glaring omission (unless I missed it) is the probabilistic programming literature. They also focus on creating programming primitives that are end-to-end differentiable and propagate distributions, similarly to the propagation of the logistic distribution here. I would expect at least some mention of the core languages, e.g. Church and other works, and how they are distinguished from the present proposal.

The section that is the least polished is between lines 150 and 191 ("Complexity and Merging of Paths"). Many parts were hard to follow, although in the end they made sense. It seems that a diagram illustrating the branching, nesting and merging of paths, annotated with example probabilities or logistic function visualizations, would aid understanding -- the text should make these concepts as concrete as possible. The 2nd paragraph frames the issue as one of perturbing inputs vs perturbing intermediate variables, which is one aspect, but another side of it is modeling *dependencies* between variables vs considering them as independent (as in the example of "x < x" where x is sampled twice). The proposal is to model variables' distributions independently, rather than their joint distribution through a collection of path traces as is common in other works / Monte Carlo approaches. A parallel to independent models such as Naive Bayes is appropriate. I did not understand the argument in line 177 about optimizing an algorithm w.r.t. runtime leading to interpolations.

Similarly, the comparison of categorical probability distributions in lines 195-201 is also not well justified; it's also not clear where exactly it is used in experiments. Lines 217-222 would benefit from a formula to make the convolution and normalization process unambiguous; in general this description is too high level. The point about aliasing effects, for example, was not clear. Finally, the convolution with a logistic filter is presented as a major difference w.r.t. other interpolation strategies -- however, the mentioned strategies (bilinear and bicubic) also correspond to well-defined convolution filters (e.g. triangular for bilinear). This needs to be highlighted -- the similarities, not the differences. The main difference would be that the logistic filter has longer tails and has potentially infinite support, while the other two filters are compact, so will not propagate gradients for far away cells. All of this should be discussed more thoroughly.

Another part where words were used instead of a formula, creating unnecessary ambiguity, is in line 251; the description of "we use this variable as a loss function" is puzzling.

The proposal for the equality operator (Eq. 9), based on a probability ratio, jumps out as not following from the initial assumptions and comes across as a bit more heuristic. What is the justification for this ratio? This squared hyperbolic secant function is actually the logistic distribution's PDF, so there may be some support there if this is fleshed out.

However, the authors could simply compose the tools they already developed to express equality. Since a==b is equivalent to a<=b & a>=b and so to not(a<b)&not(a<b), the probability of equality under the logistic distribution would be simply sigma(b-a)*sigma(a-b). This should have a similar shape to Eq. 9, but does not invoke new assumptions; it just reuses existing machinery.

The experiments are adequate, well executed and have some variety; but at least one large-scale experiment would be appreciated to show that this strategy can cope with more realistic scenarios. For example, training a RNN (e.g. on a NLP task) with a stopping criterion would represent a simple application of the conditional branching, which is less complex than the tested algorithms, but at the same time it would showcase its integration with a much larger-scale model and dataset. Another missing aspect is timing -- by avoiding expanding all paths, the proposal should do better than MC-based alternatives; this should be demonstrated with a fair time comparison.

Some technical questions: In Eq. 7, how are infinite loops dealt with? The probability of looping could never vanish (e.g. for a constant condition). Have the authors considered automatically tuning beta, for example setting it such that the gradients have a minimum absolute magnitude?

**Time Spent Reviewing:**

3

---

> ### Author Response · Authors · 2021-08-10
> **Response**
>
> Thank you for your helpful and encouraging comments.
>
> **Diagram for "Complexity and Merging of Paths":** This is a good suggestion, we propose to include a diagram to clarify it.
>
> **Differentiable wrt. runtime optimizations (line 177):**
> We agree that this is an ambiguous wording.
> What we meant is the case of improving a conventional algorithm's runtime, e.g., by using complex heuristics to reduce the runtime (for an example, see  "Corner Cases / Failure modes" below).
> If there is a relaxation of the complex heuristics that (conventionally) reduce the runtime, two problems occur:
> 1. The relaxed version of the conventionally faster algorithm is much slower because additional relaxation(s) is/are necessary and there are more paths.
> 2. Uncertainties wrt. those runtime heuristics and additional paths are introduced and gradients wrt. these will be computed. However, those gradients are not meaningful.
>
> What we specifically meant is the second point: The differentiation wrt. runtime heuristics or additional branching is usually not meaningful for algorithmic supervision.
> We will improve the discussion of this in the revision.
>
>
> **Comparison of Categorical Probability Distributions:**
> We used this in the Levenshtein distance experiments and will include this information in the revision.
>
>
> **Relaxed Indexing (lines 217-222):**
> We will formalize it mathematically it in the revision and rework its discussion.
>
> **"we use this variable as a loss function" is puzzling:**
> We agree that the wording is ambiguous and will formalize it mathematically in the revision.
>
> **Derivation for Relaxed Equality (Eq. 9):**
> Yes, all three possible derivations are (up to constants) equivalent. We propose adding the alternative derivations to the supplementary material and discuss it in detail.
>
> **Large-scale experiment:** We agree that NLP/RNN tasks are a great direction for future research.
> So far, in the larger experiments, such as the shortest path setting, we use the architecture defined for this benchmark, which is a ResNet, as an example for a large-scale architecture.
> We will emphasize this in the revision.
>
> **Minor Comments:**
>
> * We deal with infinite loops by setting a threshold for the maximum number of iterations and print a warning if it is reached. If the algorithm is correct and halts, this usually means that $\beta$ is not chosen properly.
>
> * We have not considered automatically tuning $\beta$ via the magnitude of the gradients, but we believe this to be a good idea for future research.
>
> * We will include probabilistic programming in the related work section.
>
>
> **Corner Cases / Failure modes:**
> An example for a failure mode would be implementing a relaxed Dijkstra algorithm.
> This is because for a relaxation of Dijkstra, we would interpolate between different orders of visiting nodes.
> The orders of visiting nodes are based on a heuristic, which is what makes Dijkstra fast.
> However, if we have to run all paths anyways (to compute the relaxation), it can lead to a combinatorial explosion.
> In addition, by interpolating between those alternative orders, a large amount of uncertainty is introduced.
> This uncertainty will also decrease performance.
> Therefore, it is beneficial to use an algorithm with a concrete execution structure from both a runtime and quality perspective.
>
> We will extend the discussion in the paper.

---

> > ### Comment · Reviewer_1ez2 · 2021-08-24
> > **Some missing answers**
> >
> > Thank you for your responses. I appreciate the detailed explanations, which make sense.
> >
> > However, there are a few other points -- "Comparison of Categorical Probability Distributions", "Relaxed Indexing (lines 217-222)", "we use this variable as a loss function is puzzling" --  where text improvements are promised, but no specific details are given.
> >
> > To be confident that the final paper will address these issues correctly, I'd like to know what are the alternative explanations/justifications that will be included in the paper; since they are not trivial changes, there is a non-negligible chance of introducing some mistake.
> >
> > I'd also like to know how this proposal differs from standard probabilistic programming, since it has not been discussed.
> >
> > Finally, this point was not addressed: "Another missing aspect is timing -- by avoiding expanding all paths, the proposal should do better than MC-based alternatives; this should be demonstrated with a fair time comparison." Thank you.

---

> > > ### Author Response · Authors · 2021-08-26
> > > **Further responses.**
> > >
> > > Thank you for your response!
> > > Please let us know whether you have further suggestions and questions.
> > >
> > > **Comparison of Categorical Probability Distributions**
> > >
> > > We will append the line "An example for the application of comparing categorical probability distributions is shown in the Levenshtein distance experiments in Section 4.4." to line 201. Please let us know if you think any further clarification might be helpful here.
> > >
> > >
> > > **Relaxed Indexing (lines 217-222)**
> > >
> > > We will add the following explanation in line 217-222:
> > > "The convolution of a tensor A with a logistic filter g (not to be confused with discrete-discrete convolution in neural networks) yields a function that is evaluated at point $i$. For mathematical conciseness, we denote this  in terms of a convolution. We choose the logistic filter over bilinear and bicubic filters because we model logistic distributions, and additionally because bilinear and bicubic filters have compact support whereas the logistic filter provides infinite support. This allows modeling relationships beyond the next neighbors and is more flexible as the steepness $\beta$ can be tuned for the respective application."
> > >
> > > We agree that bilinear and bicubic can also be defined via convolution. However, as relaxed indexing can also be defined without convolution, we consider this only a technicality, the convolution is not the important aspect. We will clarify that both can be written in terms of convolutions.
> > >
> > > For our normalization step, instead of computing $(g * \mathbf{A})(i) = \sum_j g(j-i) \mathbf{A}_j$, we compute $\sum_j g(j-i) \mathbf{A}_j / (\sum_j g(j-i))$ where $j$ are all valid indices for the tensor $\mathbf{A}$. We will include these equations in the paper.
> > >
> > >
> > > **Loss function for sorting supervision.**
> > >
> > > We will change the sentence to the following description:
> > > "We use the probability that the predictions are sorted correctly (under perturbation of variables) as the loss function. This equals the probability that no swap operation was necessary, and thus $\mathcal{L} = 1 - \prod_{p\in P} (1-p)$ for probabilities $p$ of each potential swap $p\in P$."
> > >
> > >
> > > **Relation to Probabilistic Programming**
> > > The goal of probabilistic programming languages (like pyro or Stan) is to infer the posterior in a Bayesian model, e.g., in a Bayesian neural network, one puts priors on the weights, and then wants to learn their posterior from the data.
> > > While probabilistic programming performs inference of a predefined probabilistic model, our work's goal is to provide gradients for learning by relaxing algorithms.
> > > We do not see a direct connection of our work to probabilistic programming.
> > > We would be happy to discuss more, if you could hint us at the specific similarities that you see between our work and probabilistic programming.
> > >
> > >
> > > **"Another missing aspect is timing -- by avoiding expanding all paths, the proposal should do better than MC-based alternatives; this should be demonstrated with a fair time comparison."**
> > >
> > > MC-based alternatives randomly sample from the potential paths.
> > > For a fair time comparison, note that the runtime of MC-based methods depends linearly on the number of samples, and the number of samples greatly impacts the quality of gradients.
> > > The adequate number of samples depends on the problem, the algorithm, and other hyperparameters such as learning rate, number of epochs, batch size, and temperature / steepness.
> > > This makes a fair time comparison non-trivial, and would also not translate to other applications / settings.
> > > Comparing runtimes between our method and perturbed optimizers on the shortest-path task, we expect the overall training time to be approximately equal because (at least for our method) the majority of the training time is spent computing the neural network outputs.
> > > In fact, except for the differentiable renderer, the runtime of the upstream neural network is always much larger than the runtime of the relaxed algorithm.

---

> > > > ### Comment · Reviewer_1ez2 · 2021-08-30
> > > > **Final score**
> > > >
> > > > Thank you, this does answer all of my points satisfactorily (especially the revised text), and I'm convinced that it is reasonable to leave out timing experiments and probabilistic programming. I will maintain my recommendation to accept the paper.

---

### Official Review · Reviewer_GwAs · 2021-07-26

**Rating:** 6
**Confidence:** 3

**Summary:**

This paper tries to integrate algorithms into end-to-end trainable neural networks.
Its main ingredient is `general approximation of discrete conditions` so that the conditions in control structures can be made differentiable. The relaxation is mostly done by perturbation from logistic distributions.

The authors evaluate the proposed method on four tasks.
1. Sorting concatenated MNIST images
2. Finding the shortest path
3. Reconstructing voxels from silhouettes
4. Computing Levenshtein distance

**Limitations And Societal Impact:**

Corner cases where the proposed method cannot handle should be mentioned.

Societal impact: n/a

**Main Review:**

> This submission is out of my expertise. I suppose it came to me regarding weak supervision but I am not familiar with differentiable algorithms. I asked the AC to remove me from the reviewers but.. no response. Anyway, I will try to review this submission to my understanding.

originality:
- The authors propose continuous relaxations of different algorithmic concepts: comparators, conditional statements, loops, and indexing.
  - The proposed method estimates expectation by sampling-free closed form solution.
- I cannot consider originality due to my lack of expertise.

quality:
- Eq. 1-5 supports choosing logistic distributions for relaxation.
- Relaxed comparators are reasonable (Eq. 8-10).
- Relaxed indexing is somewhat vague because not every array has semantic relationship between neighboring elements (L227-229).
  - The latter case should be covered in the experiments or the reason for not covering should be described.

clarity:
- Most parts are clear to understand.

significance:
- Although the proposed method has some pitfalls (e.g., x<x evaluates to 50% and the relaxed indexing is meaningful only if the array is ordered), I suppose providing sampling-free closed form solution is an important contribution to the literature.


**Time Spent Reviewing:**

2

---

> ### Author Response · Authors · 2021-08-10
> **Response**
>
> Thank you for reviewing this paper and for your helpful and encouraging comments.
>
> **Relaxed indexing and semantic orders:**
> Yes, it is correct that real-valued indexing (the one also shown in Figure 2) requires some semantic relationship between ordering elements.
> To address this point, we distinguish in the paper between real-valued indexing and marginal indexing.
> For marginal / categorical indexing, we use a categorical probability distribution produced via softmax, i.e., the maximum under logistic perturbation.
> This variant does not assume any semantic relationship between neighbors.
> We will clarify the notation and extend the discussion (which for categorical distributions was quite short.)
> We also propose including it in Figure 2 as real-valued and categorical indexing are equally important.
>
> **Corner Cases / Failure modes:**
> An example for a failure mode would be implementing a relaxed Dijkstra algorithm.
> This is because for a relaxation of Dijkstra, we would interpolate between different orders of visiting nodes.
> The orders of visiting nodes are based on a heuristic, which is what makes Dijkstra fast.
> However, if we have to run all paths anyways (to compute the relaxation), it can lead to a combinatorial explosion.
> In addition, by interpolating between those alternative orders, a large amount of uncertainty is introduced.
> This uncertainty will also decrease performance.
> Therefore, it is beneficial to use an algorithm with a concrete execution structure from both a runtime and quality perspective.
>
> We will extend the discussion in the paper.

---

> > ### Comment · Reviewer_GwAs · 2021-08-25
> > **Thanks**
> >
> > The response addresses some of my concerns and I feel positive about this paper.
> > However, I am keeping my score because I have not found enough ground for assessing the originality and significance of this work.

---

### Decision · Program_Chairs · 2021-09-27

**Decision:**

Accept (Poster)

**Comment:**

In this paper, the authors propose an approach for smoothing algorithms. While existing works were typically specialised for specific tasks, this works attempts to be more general by smoothing low-level operations such as if-else branches, while loops, and array indexing. The authors successfully demonstrate their approach on four diverse algorithms : bubble sort, shortest path, rendering and levenshtein.

The paper received an average score of 6.25, which is slightly above acceptance and one of the reviewers is ready to back up the paper.

While the paper would benefit from more polishing, it is overall well written and we would therefore like to recommend acceptance, assuming that the the authors will comply to the following requests, to further improve the paper:

- While the authors did a pretty good with citing the relevant literature, one reviewer mentioned additional references on stick breaking processes and array indexing. It is important to add these references.

- The authors should discuss more the pros and cons of Monte-Carlo approaches vs. the proposed approach. Monte-Carlo approaches, such as the one of Berthet et al, can run the original program unmodified, which is a key benefit.

- Limitations: most importantly, the authors should add a section on limitations of the approach. Indeed, while the approach is general at first glance, it has restrictions. First, the authors should discuss computational cost in more depth. Because of the local averaging, structures like trees would take exponential time to evaluate since they need to be fully expanded. Second, it is not clear for what class of programs the proposed approach is gonna work. All implemented applications (bubble sort, shortest path, rendering and levenshtein) seem to essentially have a fixed computational graph. Algorithms like Dijkstra or Quick sort wouldn't work with the proposed approach because these algorithms have inherently boolean decision steps, and making these decisions probabilistic would break the algorithm. The authors should try to delineate what program characteristics are needed for the proposed approach to work. This will help people who want to use the proposed framework know the operations they can and can't do. Lastly, when a program includes smoothed while loops, it's not clear if the program is even guaranteed to halt. Some further clarifications are needed.